# Anti-Inflammatory Activity of Bryophytes Extracts in LPS-Stimulated RAW264.7 Murine Macrophages

**DOI:** 10.3390/molecules27061940

**Published:** 2022-03-17

**Authors:** Raíssa Volpatto Marques, Stefania Enza Sestito, Frédéric Bourgaud, Sissi Miguel, Frédéric Cailotto, Pascal Reboul, Jean-Yves Jouzeau, Sophie Rahuel-Clermont, Sandrine Boschi-Muller, Henrik Toft Simonsen, David Moulin

**Affiliations:** 1Department of Biotechnology and Biomedicine, Technical University of Denmark, Søltoft Plads 223, 2800 Kongens Lyngby, Denmark; rvolpattomarques@gmail.com; 2UMR 7365 CNRS, Ingénierie Moléculaire et Physiopathologie Articulaire IMoPA, Université de Lorraine, 54500 Vandœuvre-lès-Nancy, France; ste.sestito@gmail.com (S.E.S.); frederic.cailotto@univ-lorraine.fr (F.C.); pascal.reboul@univ-lorraine.fr (P.R.); jean-yves.jouzeau@univ-lorraine.fr (J.-Y.J.); sophie.rahuel@univ-lorraine.fr (S.R.-C.); sandrine.boschi@univ-lorraine.fr (S.B.-M.); 3Plant Advanced Technologies, 19 Avenue de la Forêt de Haye, 54500 Vandœuvre-lès-Nancy, France; frederic.bourgaud@plantadvanced.com; 4Cellengo, 19 Avenue de la Forêt de Haye, 54500 Vandœuvre-lès-Nancy, France; sissi.miguel@plantadvanced.com

**Keywords:** bryophytes, mosses, *Dicranum majus*, *Thuidium delicatulum*, anti-inflammatory activity, nitric oxide

## Abstract

Bryophytes produce rare and bioactive compounds with a broad range of therapeutic potential, and many species are reported in ethnomedicinal uses. However, only a few studies have investigated their potential as natural anti-inflammatory drug candidate compounds. The present study investigates the anti-inflammatory effects of thirty-two species of bryophytes, including mosses and liverworts, on Raw 264.7 murine macrophages stimulated with lipopolysaccharide (LPS) or recombinant human peroxiredoxin (hPrx1). The 70% ethanol extracts of bryophytes were screened for their potential to reduce the production of nitric oxide (NO), an important pro-inflammatory mediator. Among the analyzed extracts, two moss species significantly inhibited LPS-induced NO production without cytotoxic effects. The bioactive extracts of *Dicranum majus* and *Thuidium delicatulum* inhibited NO production in a concentration-dependent manner with IC_50_ values of 1.04 and 1.54 µg/mL, respectively. The crude 70% ethanol and ethyl acetate extracts were then partitioned with different solvents in increasing order of polarity (n-hexane, diethyl ether, chloroform, ethyl acetate, and n-butanol). The fractions were screened for their inhibitory effects on NO production stimulated with LPS at 1 ng/mL or 10 ng/mL. The NO production levels were significantly affected by the fractions of decreasing polarity such as n-hexane and diethyl ether ones. Therefore, the potential of these extracts to inhibit the LPS-induced NO pathway suggests their effective properties in attenuating inflammation and could represent a perspective for the development of innovative therapeutic agents.

## 1. Introduction

The medicinal use of many species of bryophytes, including mosses and liverworts, has been mostly reported in traditional Chinese, Indian, and Native American medicines [1]. Bryophytes have shown many ethnomedicinal applications such as for the treatment of skin diseases, inflammation, microbial infections, and many others [1,2]. Indeed, bryophytes produce important specialized metabolites, particularly terpenoids (mono-, sesqui-, and diterpenoids) and aromatic compounds (mainly flavonoids, (bis)bibenzyls) as well as lipids, which have shown important biological activities [3]. Although bryophytes are a valuable source of bioactive molecules, their biological properties and chemical constituents remain relatively unexplored.

Inflammation is the physiological response of the body to overcome and contain infections (microbial) and injuries (physical, chemical, etc.) [4]. Inflammatory reactions are generally acute but can become chronic leading to many diseases [5,6]. Pathogen-associated (PAMPs) and danger-associated (DAMPs) molecular pattern molecules are derived from microorganisms or released from damaged cells which are then recognized by pattern recognition receptor (PRR)-bearing cells activating the inflammatory response [7,8]. Lipopolysaccharides (LPS) are examples of PAMPs found in the outer membrane of Gram-negative bacteria [7]. In addition, peroxiredoxins (Prxs) act as alarmins and have been reported to play important roles in innate immunity by activating macrophages and promoting DAMPs-associated inflammatory diseases [9,10,11,12]. Activated macrophage cells release a wide range of inflammatory mediators including nitric oxide (NO) and pro-inflammatory cytokines such as tumor necrosis factor-α (TNF-α), interleukin (IL)-6, IL-1β, and IL-12 that are important signaling molecules in the inflammatory reaction [13]. NO is an indicator of the inflammatory response and is synthesized by three types of NO synthases (NOS) (endothelial, neuronal, and inducible NOS (iNOS)) [14]. During inflammation, increased levels of NO produced by iNOS have an important pathological role in many inflammatory diseases [15]. To evaluate the NO levels the model murine macrophage cell line RAW 264.7 was chosen. RAW 264.7 is a commonly monocyte/macrophage model often used to investigate the bioactivity of plant-derived extracts and isolated natural products. Several studies have reported the anti-inflammatory potential of different groups of plants and their metabolites by evaluating the decreasing of NO production and other inflammatory mediators in stimulated RAW 264.7 cells [16,17,18,19]. Therefore, those studies support our choice for RAW 264.7 cell line as a suitable model for anti-inflammatory screening studies.

In this study, the 70% ethanol extracts of thirty-two species of bryophytes, including mosses and liverworts, were evaluated for attenuating the NO production induced by LPS and hPrx1 molecules. The bioactive extracts were further partitioned into non-polar to polar fractions and tested for their nitric oxide inhibitory activity. We demonstrate that extracts and fractions of the mosses *Dicranum majus* and *Thuidium delicatulum* (Bryophyta) exhibited significant inhibitory effects on NO production in LPS-induced RAW 264.7 cells.

## 2. Results and Discussion

### 2.1. Effect of Extracts on the Viability of RAW 264.7 Murine Macrophage Cells

Before performing the activity screening, metabolic effects and, consequently, cytotoxicity of 70% ethanol extracts from bryophytes, were evaluated at 100 µg/mL on RAW 264.7 macrophage cells by MTT assay, setting the threshold of cell viability at 70% (Figure 1). The extracts did not show cytotoxicity at the indicated concentration.

### 2.2. Anti-Inflammatory Effects of Extracts in hPrx1 or LPS Stimulated RAW264.7 Murine Macrophage Cells

Nitric Oxide (NO) is a typical marker of inflammation produced in response to a pathogen as well as DAMP, such as hPrx1 [12]. Intracellular hPrx1 is a peroxidase involved in the redox signaling in physiological conditions, but it was proposed to function as DAMP by activating Toll-like receptor (TLR) 4. Diverse stress conditions, including cerebral ischemia [12,20,21], induce the release of hPrx1 in the extracellular environment with increased expression of TLR4, nuclear translocation of nuclear factor κB (NF-κB) p65, and production of pro-inflammatory mediators (NO, TNF-α and IL-6). The anti-inflammatory activity of 70% ethanol extracts of a range of bryophytes was determined by assessing their potential to inhibit the production of NO induced by hPrx1 in RAW cells (Figure 2). Cells were stimulated with 300 nM hPrx1, and after 18 h incubation, nitrite quantification was used as an indicator to estimate the level of NO-induced hPrx1 in the medium by the Griess reagent method (Figure 2A). The inhibitory activity of the extracts on NO production was analyzed by pre-treating the cells with 100 µg/mL of 70% ethanol extracts for 1 h and then stimulating with hPrx1 (300 nM) for 18 h. The obtained NO levels were normalized on hPrx1 (Figure 2B). However, none of the tested whole extracts was considered bioactive in these conditions.

Sterile inflammation is not the unique responsible for NO production in cells, but bacterial endotoxins can also induce strong and wide immune responses. Thus, the anti-inflammatory effect of 70% ethanol extracts in reducing NO level was evaluated by performing Griess reaction on supernatants of lipopolysaccharides (LPS)-stimulated RAW cells (Figure 3). Cells were stimulated with 1 ng/mL LPS from *Salmonella typhimurium* for 18 h, and after incubation, the level of NO-induced LPS was quantified by Griess reaction (Figure 3A). The cells were pre-treated with 100 µg/mL of 70% ethanol extracts for 1 h and then stimulated for 18h with 1 ng/mL LPS. The NO content of the culture medium was analyzed by the Griess reagent method and NO levels normalized on LPS. Among the tested samples, the extracts of the mosses *Dicranum majus* and *Thuidium delicatulum* significantly inhibited the production of NO in LPS-stimulated cells, at 68% and 41%, respectively (Figure 3B).

The bioactive extracts were tested again at increasing concentrations finding that they decrease the NO level in a concentration-dependent manner with IC_50_ values of 1.04 µg/mL and 1.54 µg/mL for *D. majus* and *T. delicatulum* extracts, respectively (Figure 4).

### 2.3. Anti-Inflammatory Effects of Fractionated Extracts in LPS-Stimulated RAW264.7 Murine Macrophage Cells

The dried 70% ethanol extracts of *D. majus* and *T. delicatulum* were further partitioned into a gradient of non-polar to polar fractions through a series of organic solvents (n-hexane, diethyl ether, chloroform, ethyl acetate, and n-butanol). The whole extracts and fractions at 10 µg/mL showed no cytotoxicity on RAW264.7 cells, as indicated by Appendix A. The fractions at 10 µg/mL were then screened for their inhibitory effects on NO production in RAW 264.7 cells stimulated with LPS at 1 ng/mL or 10 ng/mL for 18 h. The fractions obtained for solvents of low polarity, such as n-hexane and diethyl ether, showed the maximum effect of NO reduction induced with 1 ng/mL and 10 ng/mL doses (Figure 5A,B). For both mosses, the inhibitory activity of those fractions on LPS (10 ng/mL)-induced NO was found to be higher than the 70% ethanol crude extracts. The inhibition of n-hexane fractions on LPS (10 ng/mL)-induced NO was observed to be 78% and 66% for *D. majus* and *T. delicatulum*, respectively (Figure 5A,B). In addition, the diethyl ether fractions of *D. majus* and *T. delicatulum* showed 53% and 52% of NO inhibition, respectively.

The bioactive plants were subsequently extracted with ethyl acetate and both crude extracts and derived fractions (n-hexane, diethyl ether, chloroform, ethyl acetate, and n-butanol) were screened for NO inhibition stimulated with LPS at 1 ng/mL or 10 ng/mL for 18h. As above, the whole extracts and fractions at 10 µg/mL showed no cytotoxicity on RAW264.7 cells (Appendix A).

The anti-inflammatory results indicated that the inhibitory effect of *D. majus* was prevalent in the fractions prepared with less polar solvents (Figure 6A). The inhibition of n-hexane and diethyl ether fractions of *D. majus* on LPS (10 ng/mL)-induced NO was observed to be 65% (Figure 6A). In addition, the ethyl acetate crude extract of *D. majus* exhibited the most potent LPS (10 ng/mL)-induced NO inhibition (60%) as compared to 70% ethanol crude extract (20%, Figure 5A). In general, *T. delicatulum* ethyl acetate crude extracts and obtained fractions exhibited lower potential to decrease NO production in both induced LPS 1 and 10 ng/mL doses. Among the tested fractions, the NO was decreased below 36% in LPS (10 ng/mL)-stimulated cells (Figure 6B). There was no pronounced difference in the NO inhibition potential among the fractions derived of ethyl acetate crude extract on LPS (10 ng/mL)-induced cells, as indicated by the percent inhibition of ethyl acetate crude extract (29%), n-hexane (33%) and diethyl ether (26%) fractions.

The results indicate the presence of bioactive constituents in both crude extracts and fractions possessing significant anti-inflammatory activity. Hence, it was observed that the non-polar fractions had a higher anti-inflammatory activity. Those different levels of activity may be related to the concentration of bioactive compounds in the fractions as well as their polarity.

Other studies explored the anti-inflammatory activity of extracts from bryophytes species on NO production inhibition. Thereby, the treatment with 50 μg/mL of peat moss (*Sphagnum* sp.) aqueous extract inhibited the production of NO in LPS-stimulated (500 ng/mL, 24 h) RAW 264.7 cells [22]. Nevertheless, the extracts of *Sphagnum teres* and *Sphagnum fimbriatum* (Bryophyta) analyzed in the present study showed no anti-inflammatory activities. The methanol extract of *Polytrichum commune* (Bryophyta) was reported to inhibit the NO production induced by the treatment of LPS (1 µg/mL, for 24 h) with an IC_50_ of 65.15 µg/mL [23]. However, the extract from *Polytrichum formosum* (Bryophyta) showed no anti-inflammatory activities in our screening. Differences in anti-inflammatory properties between plant species may be due to their different chemical constituents, which may also vary depending on the species’ geographical origin and exposure to various environmental factors (e.g., season, soil, climate, etc.) [24,25].

In previous studies, *D. majus* has been reported with antibacterial activity [26]. The dichloromethane extract from *Dicranum scoparium* (Bryophyta), which belongs to the same genus of *D. majus*, was reported with anti-inflammatory activity by inhibiting 90% of 15-lipoxygenase (15-LOX) at 100 µg/mL, which has an important role in the inflammatory pathway [27,28]. Investigations of the biological properties of the ethanol extract and fractions from *T. delicatulum* were described with antibacterial and antifungal activities as well [29]. *Thuidium* spp. have been reported as antibacterial and anti-inflammatory agents in China [30]. Moreover, a terpenoid-rich fraction of the methanol crude extract of *Thuidium tamariscellum* (Bryophyta) has exhibited anti-inflammatory activities by inhibiting the activity of enzymes involved in inflammatory pathways such as cyclooxygenase, LOX, and myeloperoxidase and also by decreasing the levels of LPS-induced NO [31].

## 3. Material and Methods

### 3.1. Plant Material

Thirty specimens of bryophytes were collected from different locations, including Germany, Denmark, Sweden, and Iceland. Three other species of mosses were purchased from a moss provider (Bryoflor, Paris, France) (http://www.bryoflor.com/, accessed on 20 February 2019). The list of species is found in the Appendix A. The specimens were identified by Professor Dr. Nils Cronberg (Department of Biology, Faculty of Sciences, Lund University, Lund, Sweden). Voucher specimens of *Dicranum majus* (ID no MTRaMa30) and *Thuidium delicatulum* (ID no MTRaMa34) were sent for deposition at the Lund University Botanical Museum (LD). The whole plants were dried at room temperature (samples from Germany) or in an oven at 40 °C and ground to a fine powder using a bead mill.

### 3.2. Recombinant hPrx1 Production

Recombinant wild-type hPrx1 was produced in *Escherichia coli* as an N-terminal fusion with a His-tag. hPrx1 was produced and purified as described by Kriznik et al. [32]. LPS contamination of purified Prx was checked by stimulating cells with denatured protein (95 °C, 5 min). As a control heated LPS was tested in the same experiment. NO was detected only in samples treated with heated LPS and not in heat-denatured Prx.

### 3.3. Extraction of Small Molecules for Screening Activities

The powdered plants were homogenized in 70% ethanol (*v/v*) in water (1:10 g/mL of dry weight to solvent ratio) for the extraction of small molecules. The extractions were performed in an ultrasound bath (Blacksonic 275H) at 40 °C for 30 min followed by 24 h in an agitation mixer, adapted from previously reported protocol [33]. *Polytrichum formosum* (Bryophyta) and *Bazzania trilobata* (Marchantiophyta) were extracted by maceration for 30 min in a rotating mixer at room temperature, adapted from previously reported protocol [34]. After centrifugation, the supernatant was collected and used for analysis. The 70% ethanolic extracts were diluted to the concentrations as indicated in each experiment.

### 3.4. Extraction and Fractionation of Bioactive Bryophytes

According to the results of the screenings, the bioactive plant extracts were selected for further analysis. A second round of extractions was performed in which the powdered plants were homogenized in 70% ethanol (*v*/*v*) in water or ethyl acetate (1:10 g/mL of dry weight to solvent ratio) for the extraction of small molecules as previously described. After extractions, the solvents were evaporated under vacuum conditions. The remaining dry crude extracts of 70% ethanol and ethyl acetate extracts were suspended in water and successively partitioned with n-hexane, diethyl ether, chloroform, ethyl acetate, and n-butanol (Table 1). The crude 70% ethanol and ethyl acetate extracts and their derivative fractions were evaporated and suspended in ethanol absolute for analysis.

### 3.5. Cell Culture

Murine macrophages Raw 264.7 cells were cultured in Dulbecco’s Modified Eagle’s Medium (DMEM, Gibco Fisher Scientific) high glucose supplemented with 10% fetal bovine serum, Penstrep 1X, glutamine (2 mM), HEPES 20 mM pH 7.3, at 37 °C, 5% CO_2_, 95% humidity. Cells were washed in warm PBS, detached using a cell scraper and the cell number was estimated using Trypan Blue (Sigma-Aldrich, St. Louis, MO, USA).

### 3.6. Cell Viability and Cytotoxicity Assay

Raw 264.7 cells were seeded in a 96-well plate in 100 μL of DMEM without Phenol Red at a density of 1 × 104 cells/well and incubated overnight at 37 °C, 5% CO_2_, 95% humidity. Then, the medium was removed, and the cells were treated with 100 µg/mL of the 70% ethanol extracts diluted in complete DMEM (90 µL/well, final solvent concentration = 0.1%). DMSO 10% and PBS 10% were included as negative and positive controls, respectively. After overnight incubation, 10 μL of MTT solution (5 mg/mL in PBS) were added to each well. After 3 h incubation (37 °C, 5% CO_2_, 95% humidity), formazan crystals were dissolved with 100 µL/well of HCl 0.1 N in 2-propanol. Formazan concentration was determined by measuring the absorbance at 570 nm (Varioskan, Thermofisher, Waltham, MA, USA). The results were normalized on untreated control (PBS) and expressed as the mean of percentage ± standard deviation of two independent experiments (n = 3–6).

The whole bioactive extracts and derived fractions at 10 µg/mL were also tested using MTT cell viability test and the Lactate Dehydrogenase (LDH) assay to validate cell death (%) and cytotoxicity. The additional data is available in the Appendix A. For the LDH assay, the RAW 264.7 cells were inoculated into a complete culture medium in 96-well plates, at a rate of 10^4^ cells per well. After incubation at 37 °C overnight, the medium was renewed with DMEM 4.5 g/L culture medium without pyruvate with 0.1% Fetal Bovine Serum (FBS) containing penicillin/streptomycin and glutamine, incubated at 37 °C overnight. The medium was aspirated again, and the cells were treated with 10 μg/mL of extracts/fractions, which were diluted in the medium used previously. After incubation overnight at 37 °C, the released LDH was assayed using the Cytotoxicity Detection Kit LDH assay (Roche) which contains the catalyst and the solution for lysis of the cells. The lysis solution was added to the cells which have not been previously treated (7.5 μL of lysis solution for a volume of 150 μL), then incubated for 15 min at 37 °C (these wells represent the positive controls of this test). Positive control (no treated lyzed cells) and negative control (0.1% DMSO). Then, 100 μL of supernatant from each well were recovered and moved to a new 96-well plate and 100 μL of mix containing the catalyst were then added to each well. The plate was incubated in the dark for about 20 to 30 min, then the reaction was blocked by adding 50 μL per well of stop solution. Absorbance was measured with a plate reader (Varioskan) at a wavelength of 490 nm. The results are expressed as the mean of percentage ± standard deviation (n = 2–3).

The formula that was used to calculate the percentage of cytotoxicity is as follows:(1)$$\frac{Sample\;absorbance-lowest\;control\;absorbance\;}{Highest\;control\;absorbance-lowest\;control\;absorbance\;}\times 100$$

### 3.7. Measurement of Nitric Oxide

Raw 264.7 cells (5–7 × 10^5^ cells/well) were seeded in 96-well plates in 150 μL of complete DMEM and incubated for 24 h. Then, the medium was removed and the cells were treated with 100 µg/mL of crude extracts (for the first screening) or 10 µg/mL of bioactive extracts/fractions (190 µL/well), in triplicate. One hour later, the cells were stimulated with 1 ng/mL or 10 ng/mL LPS from *Salmonella typhimurium* or 300 nM of hPrx1 (10 µL/well) for 18h. The supernatants were collected and the Griess reaction was performed for nitrite quantification. The plate reading was assessed by using a spectrophotometer at 540 nm. Nitrite quantification was estimated by interpolating the standard curve and then normalized on untreated and stimulated cells. The bioactive extracts were tested in dose-response by applying the same test.

### 3.8. Statistical Analysis

Statistical analyses were performed using Prism Software (GraphPad Software, Inc., La Jolla, CA, USA). Differences between the mean values were assessed by one-way analysis of variance (ANOVA) followed by Bonferroni multiple comparison test. *p* < 0.05 was considered statistically significant. The IC_50_ values were determined by non-linear regression analysis (GraphPad Prism software version 9).

## 4. Conclusions

The present study investigated the anti-inflammatory effects of thirty-two species of bryophytes on human peroxiredoxin (hPrx1) and lipopolysaccharide (LPS) stimulated RAW264.7 murine macrophage cells. The 70% ethanol extracts were screened for their potential to reduce the production of nitric oxide (NO). Although the whole extracts showed no inhibition of NO stimulated by hPrx1, two species of mosses significantly inhibited LPS-induced NO. The bioactive extracts of *D. majus* and *T. delicatulum* inhibited NO in a concentration-dependent manner with IC_50_ values of 1.04 and 1.54 µg/mL, respectively. Among the tested fractions of the crude extracts, the n-hexane and diethyl ether fractions reduced NO production more efficiently. The potential of the extracts to inhibit LPS-induced NO pathway indicates their effective properties in attenuating the inflammatory response. The inhibitory properties of these extracts may present new sources of natural ingredients for anti-inflammatory drug discovery. Further studies on their efficacy activities, mode of actions, and identification of bioactive compounds should be performed.

## Figures and Tables

**Figure 1 molecules-27-01940-f001:**
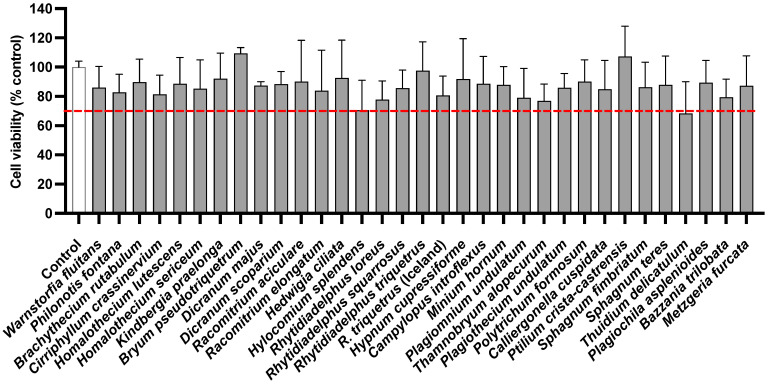
Effect of 70% ethanol extracts on the viability of RAW 264.7 cells determined by MTT assay. Cells were treated with 100 µg/mL of extracts for 24 h. Data represent mean ± standard deviation (n = 3–6) and values are normalized on control (PBS). The tested extracts revealed no significant difference with *p*-values > 0.05, calculated with ANOVA. The dashed line represents 70% of cell viability.

**Figure 2 molecules-27-01940-f002:**
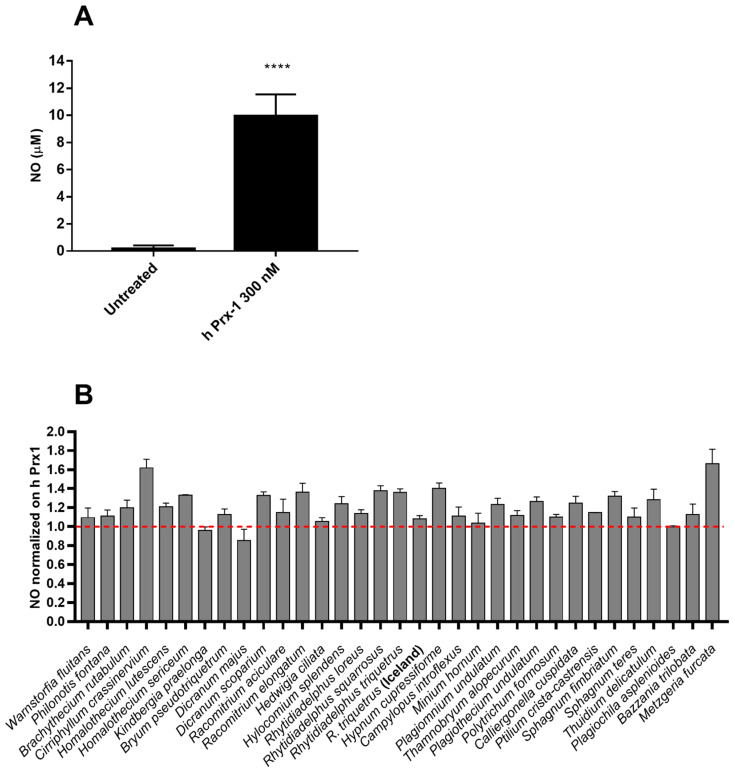
(**A**) NO-induced hPrx1. Cells were stimulated with 300 nM hPrx1 and the amount of induced NO was quantified by Griess reaction. The graph represents mean ± standard deviation; ANOVA analysis and Tukey’s test were used for the analysis (**** *p* < 0.0001). (**B**) Effect of 70% ethanol extracts on hPrx1-induced NO in RAW 264.7 cells. Cells pre-treated with 100 µg/mL of extracts for 1 h were stimulated with hPrx1 (300 nM) for 18 h. The NO content of the culture medium was analyzed by the Griess reagent method. Data represent means ± standard deviation (n = 3) and values are normalized on hPrx1. The dashed line represents the level of NO in cells stimulated by hPrx1 alone.

**Figure 3 molecules-27-01940-f003:**
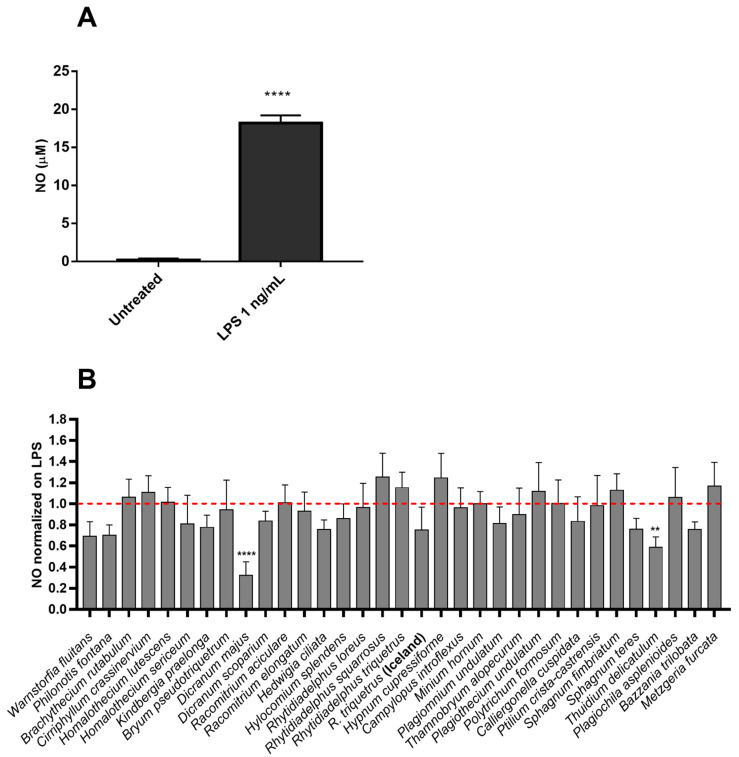
(**A**) LPS-NO induction in murine macrophages. The amount of NO in cells treated with LPS 1 ng/mL for 18 h is significantly different (**** *p* < 0.0001) compared to untreated cells. The graph represents the mean ± standard deviation of three independent experiments. ANOVA and Tukey’s test were used for statistical analysis. (**B**) Effect of 70% ethanol extracts on LPS-induced NO in RAW 264.7 cells. Cells pre-treated with 100 µg/mL of extracts for 1 h were stimulated with LPS (1 ng/mL) for 18h, which is enough to induce a significant signal. The NO content of the culture medium was analyzed by the Griess reagent method. Data represent mean ± standard deviation (n = 7–11) and the results normalized by the effect of LPS (1 ng/mL) on untreated and stimulated cells. The tested extracts revealed a significant difference with *p*-values < 0.05, calculated with ANOVA. ** *p* < 0.01, **** *p* < 0.0001 indicate significant differences compared to the control. The dashed line represents the level of NO in cells stimulated by LPS alone. The amount of nitrite in the media was calculated from sodium nitrite standard curve.

**Figure 4 molecules-27-01940-f004:**
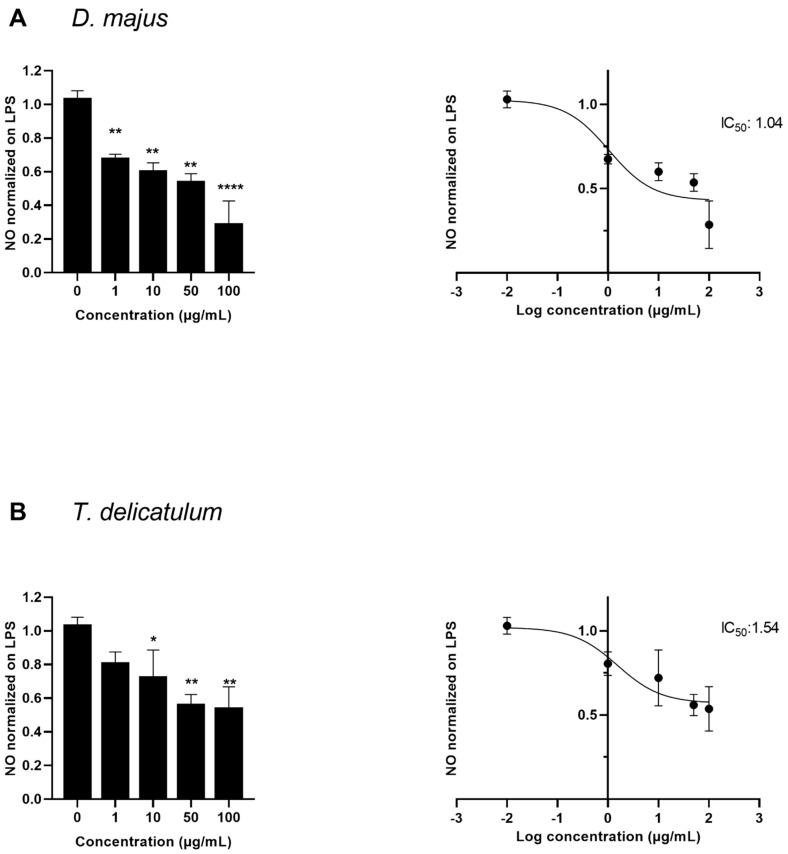
Dose-response effect and IC_50_ values of 70% ethanol extract from (**A**) *D. majus* and (**B**) *T. delicatulum*, on LPS-induced NO in RAW264.7 cells. Data represent mean ± standard deviation (n = 2–3 of cells treated with extracts) and values are normalized on LPS (1 ng/mL). * *p* < 0.05, ** *p* < 0.01, **** *p* < 0.0001 indicate significant differences compared to the control.

**Figure 5 molecules-27-01940-f005:**
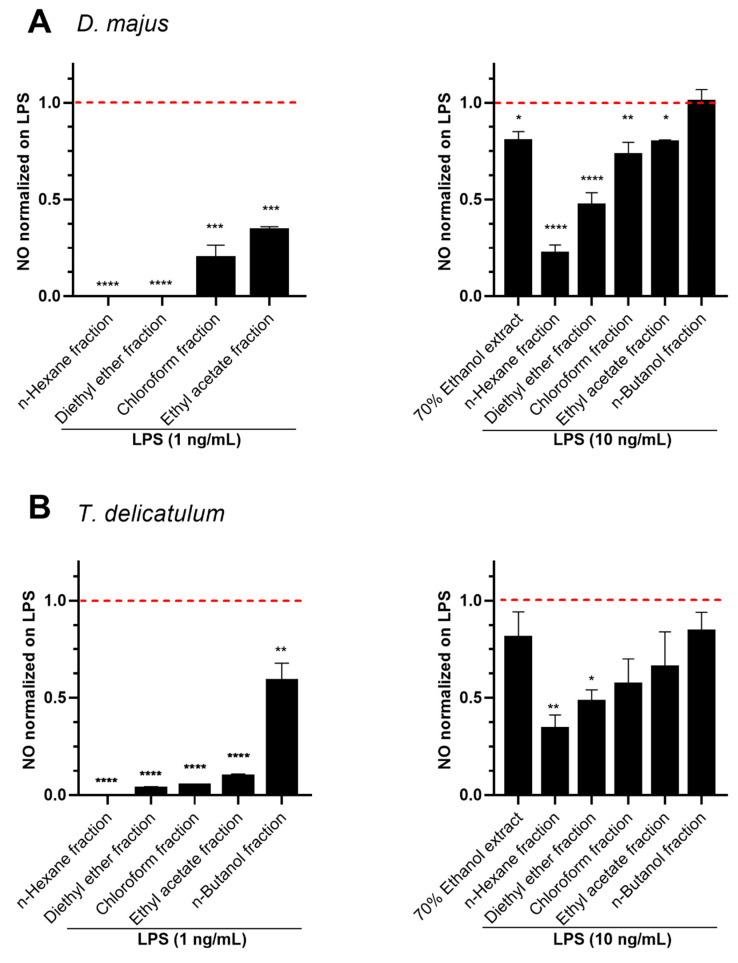
Effect of the fractions of the 70% ethanol crude extracts from (**A**) *D. majus* and (**B**) *T. delicatulum* on LPS-induced NO in RAW264.7 cells. Cells pre-treated with 10 µg/mL of extracts for 1 h were stimulated with LPS at 1 ng/mL or 10 ng/mL for 18 h. The NO content was analyzed by the Griess reagent method. Data represent mean ± standard deviation (n = 2–3). Values are normalized on LPS. The tested obtained extracts performed on the 1 and 10 ng/mL treatments, revealed a significant difference with *p*-values < 0.05, calculated with ANOVA. * *p* < 0.05, ** *p* < 0.01, *** *p* < 0.001, **** *p* < 0.0001 indicate significant differences compared to the control. The dashed line represents the level of NO in cells stimulated by LPS alone.

**Figure 6 molecules-27-01940-f006:**
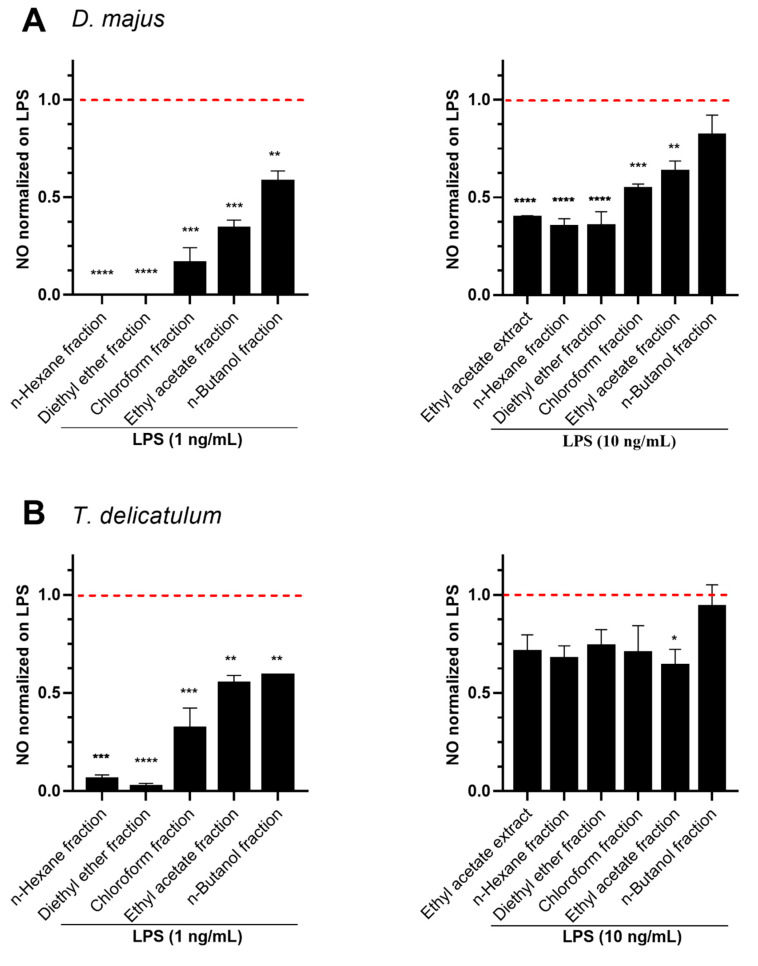
Effect of the fractions of the ethyl acetate crude extracts from (**A**) *D. majus* and (**B**) *T. delicatulum* on LPS-induced NO in RAW264.7 cells. Cells pre-treated with 10 µg/mL of extracts for 1 h were stimulated with LPS at 1 ng/mL or 10 ng/mL for 18 h. The NO content of the culture medium was analyzed by the Griess reagent method. Data represent mean ± standard deviation (n = 2–3). Values are normalized on LPS. The tested extracts revealed a significant difference with *p*-values < 0.05, calculated with ANOVA. * *p* < 0.05, ** *p* < 0.01, *** *p* < 0.001, **** *p* < 0.0001 indicate significant differences compared to the control. The dashed line represents the level of NO in cells stimulated by LPS alone.

**Table 1 molecules-27-01940-t001:** Fractions yield (%) of bioactive crude extracts.

Fractions Yield (%)	*D. majus* (70% Ethanol Crude Extract)	*D. majus* (Ethyl Acetate Crude Extract)	*T. delicatulum* (70% Ethanol Crude Extract)	*T. delicatulum* (Ethyl Acetate Crude Extract)
n-Hexane	7	13	5	8
Diethyl Ether	16	30	56	29
Chloroform	15	16	10	18
Ethyl Acetate	7	9	8	19
n-Butanol	12	10	10	23

## Data Availability

Not applicable.

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
