# Peer review of "Anti-Inflammatory Activity of Bryophytes Extracts in LPS-Stimulated RAW264.7 Murine Macrophages"

_molecules, 2022, doi:10.3390/molecules27061940_

Round 1
Reviewer 1 Report
Unfortunately, this reviewer thinks the methods used are no acceptable.
The volume of ethanol in all of the test samples is too high. The extraction techniques are referenced, but, again, the reconsitution in 100% ethanol after extraction and the use of 10% DMSO in controls is way too high. This content should be at 0.1% with 1% being acceptable with a very detailed explanation as to why it is necessary.
Also, the authors state that the whole bioactive extracts and derived fractions at 10 μg/mL were used and this is not clear. If the extracts were dissolved in absolute ethanol, how much ethanol was used to dissolve 10 μg? The experimental details are simply not robust enough for this author to suggest publication of this manuscript.
Author Response
We would like to thank the reviewer for the revision and comments.
For the experiments, we have prepared high concentrated extracts and fractions and tested them at concentrations of 100 μg/mL or 10 μg/mL (as indicated in the manuscript), so the final concentration of the solvent is 0.1%. The tests of MTT and LDH showed that none of the extracts and fractions studied were toxic at the indicated concentrations. 10% DMSO is a control of mortality (negative control of viability).
Reviewer 2 Report
The manuscript entitled "Anti‑inflammatory activity of bryophytes extracts in LPS-stimulated RAW264.7 murine macrophages" addresses a relevant and appropriate topic for this journal. The manuscript is well structured, well reasoned and well written. After including the corrections suggested below, I am of the opinion that it should be accepted for publication.
Corrections needed:
line 74 - that extracts and fractions of the mosses Dicranum majus and Thuidium delicatulum (Bryophyta) ...
line 162 - ... D. majus and T. delicatulum, respectively ... (Note: "D. majus, T. delicatulum" in italics)
line 207 - (Sphagnum sp.) aqueous extract ... (Note: "sp." is not in italics)
line 208/209 - ... Sphagnum teres and Sphagnum fimbriatum (Bryophyta) ...
line 210 - ... Polytrichum commune (Bryophyta) ...
line 212 - ... Polytrichum formosum (Bryophyta) ...
line 218 - ... Dicranum scoparium (Bryophyta) ...
line 222/223 - ... Thuidium spp. have ... (Note: "spp." is not in italics)
line 224 - ... extract of Thuidium tamariscellum (Bryophyta) ...
line 249/250 - ... Polytrichum formosum and Bazzania trilobata (Bryophyta) were extracted ...
Author Response
We would like to thank the reviewer for his/her helpful comments. These corrections are now incorporated throughout the manuscript.
Reviewer 3 Report
This is a very interesting paper demonstrating the effects of bryophyte extracts on mouse macrophages that were stimulated with LPS.
While iNOS is indeed a key inducible enzyme from LPS-stimulated macrophages, there are other biomarkers of inflammation, including IL-6, TNF-alpha, IL-8, COX-2, and others.
NO production is a key marker of M1-polarized macrophages....
but the investigators did not examine the possibility that some extracts actually polarized the macrophages to a M2 phenotype, and thus may be anti-inflammatory by another mechanism.
Nonetheless, most of the work in this study was on the chemistry side of the equation, and this was very good.
It would be good for the readership if the authors explained why they did not examine cytokines (IL-6 vs IL-10 as an example).
Author Response
We would like to thank the reviewer for his/her suggestions. We agree that our findings would benefit from further investigations; however, these studies cannot be conducted due to limitation of plant material and financial resources.
Round 2
Reviewer 1 Report
The changes by the authors to the manuscript entitled "Anti‑inflammatory activity of bryophytes extracts in LPS-stimulated RAW264.7 murine macrophages" are acknowledged and this reviewer do believe this manuscript is appropriate for publication in Molecules.